# Timing in Lower Limb Complex Movement Tests for DanceSport Athletes: Relation between FitLight Trainer and IMU Measurements

**DOI:** 10.3390/s23031456

**Published:** 2023-01-28

**Authors:** Marija Prelević, Milivoj Dopsaj, Sara Stančin

**Affiliations:** 1Faculty of Sport and Physical Education, University of Belgrade, Blagoja Parovića Street 156, 11030 Belgrade, Serbia; 2College of Sports and Health, Toše Jovanovića 11, 11030 Belgrade, Serbia; 3Faculty of Electrical Engineering, University of Ljubljana, Tržaška c. 25, 1000 Ljubljana, Slovenia

**Keywords:** sport, dance, FitLight, inertial measurement unit, wearable devices, lower limb movements, timing measurements

## Abstract

We examine the relation between two devices used in measuring the timing in lower limb complex movement tests for DanceSport athletes, an inertial measurement unit (IMU) and a FitLight Trainer device, with the latter regarded as the gold standard method in the field. Four tests are selected to cover the lower limb movements. The research sample comprises 21 experienced dancers from different dance disciplines, performing the four tests with each of their lower limbs. Compared using concurrent validity, the two devices used show great agreement for estimating the total tests’ run times, with interclass correlation coefficients between 0.967 and 0.994 for all tests. This agreement is additionally confirmed by Bland–Altman plots. As an alternative to other devices, the IMU sensor has proven to be a precise and suitable device for measuring timing and testing in sports. Its mobility, light weight, and size are advantages of this device in addition to measurement accuracy.

## 1. Introduction

Measuring complex limb motion abilities is an important subject of research in various types of sports. Abilities that are significant should be specially monitored and developed in the training process due to their influence on the results. For example, significant in DanceSport is strength as well as a greater range of motion in the hip joint. Additionally, Twitchett et al. [1] find that ballet dancers in particular can lack strength in the upper body, quadriceps, and thigh tendons and this should be accounted for in order for the individuals to develop into elite dancers.

The specificity of the training process, taking into account the specificity of different dance disciplines, determines the development of specific abilities.

Appropriate diagnostic tests and adequate methods are a constant topic of research in sports. The demands placed on athletes also require that coaches and trainers use appropriate tools, both for predicting and identifying future talents and for improving work with existing athletes.

Human motion monitoring based on commercially available sensors has been widely adopted and intensively studied in recent years due to the possibility of wide application in various fields [2]. Wearable devices that measure some physical quantity have already become part of the daily life of many individuals [2]. However, the demands in sports are more firm, aiming for greater precision, obtained with higher sampling frequencies as opposed to obtaining mainly statistical descriptions of the measured quantities that simple devices provide. Erlikh et al. [3] show that hundreds of technologies have been developed and applied to assess functional status, using software and hardware diagnostics, most of which solve specific problems and do not reflect the multiparametric integration of athletic ability for competitive activity. 

Pustišek et al. [4] present a brief introduction to motor learning in sport and the need for technological support as well as outlining the benefits and limitations of various sensors used for signal acquisition in sports activities, means of communication, and properties and limitations of the communication channels. Inertial measurement unit (IMU) devices, comprising 3D accelerometers, gyroscopes, and magnetometers due to their small size and light weight, transferability, low power consumption, and ease of application and use, enable the long-term monitoring of movements in situated environments. Marković et al. [5] show that IMU sensors are practically applicable when measuring the ability of fast hand movements of female athletes. The mentioned sensors have proven to be sensitive measurement tools that provide a reliable information base for objectifying the assessment of elite athletes [6]. Marković et al. [7] present a method that provides insight into the coordination of articulated human movements, measuring movement synchronization and event timing using IMU sensors with additional information about the studied structure of rapid discrete movements in various sports activities that are not perceptible to human senses.

Dancers benefit greatly from various sensor-based assistive technologies, which have been addressed by various authors, mainly taking into account step detection and performance tempo estimation. Aylward et al. [8] present a design of a wireless and compact sensor module that, when worn on the hands and feet, captures expressive gestures in real time in an interactive dance ensemble. Saltate [9], a wireless prototype support system for beginners in ballroom dancing, collects data from force sensors placed under the dancer’s feet by detecting steps. It compares the timing of the steps with the timing of the beats of the music to which the steps are performed. Dancing Coach [10] uses a Kinect device to assist dancers by extracting dance steps. Stančin and Tomažič [11] present a method for estimating dance tempo through the acquisition of 3D accelerometer signals using a wearable inertial device mounted on the dancer’s leg. Kinect-based systems allow the extraction and estimation of movement beat alignment using a dance video clip as input [12,13]. In [14], machine learning models with wearable sensors are presented. These models provide a field-based system for estimating ground reaction force during ballet jumps.

FitLight Trainer (Sport Corp., Ontario, Canada) [15] is a commercial system relying on eight LED lights and touch and motion sensors. All are integrated into wireless units 10 cm in diameter and weighting 0.3 kg. These units are controlled through a dedicated tablet device with a wireless range of 75 m and are used to guide an athlete during his or her training session. Athletes can perform tests of a pre-planned motion structure. Direct comparison between tests is another advantage of this system. Using a FitLight Trainer, Rauter et al. [15] examine different levels of sports performance and present a useful and efficient tool for its development through the improvement of motor abilities. In particular, the authors observe reactive agility through randomly selected movement stimulus and speed of movement direction change. The participants were young physical education students and football players. The FitLight Trainer device has proven to be a reliable measuring device for analyzing the relationship between simple and complex reaction times [16,17].

The goal of this research is to compare two different measurement devices, the commercially available FitLight Trainer on the one hand, and a wearable IMU on the other. Both devices are compared with respect to the performance timing they measure in complex lower limb movement tests, specially designed for dancers. Our goal is to answer the question of whether an IMU sensor can provide valid and reliable test timings compared to the results obtained with a FitLight Trainer device that is already being broadly used for this purpose in the community. 

## 2. Materials and Methods

### 2.1. Research Sample

In total, 21 experienced dancers from different dance disciplines, including Latin and Standard dance, Modern and Contemporary dance, Hip-Hop and dance section (social dances), participated in the study. Twelve participants were women (age: 19.9 ± 4.6 years; body height: 169.0 ± 4.6 cm; body weight: 58.8 ± 5.2 kg; training experience: 8.8 ± 3.1 years) and nine were men (age: 20.4 ± 4.4 years; body height: 183.4 ± 8.1 cm; body weight: 74.8 ± 9.2 kg; training experience: 9.3 ± 5.0 years). None of the participants were injured at the time of the study. They were informed about the purpose of the research as well as the procedures, and they signed a written consent form. For individuals under 18 years of age, parental consent was obtained. The study was approved by the Ethics Committee of the University of Belgrade, Faculty of Sport and Physical Education (02 No. 484-2), following the guidelines of the Declaration of Helsinki. 

### 2.2. Equipment

A standard 3 cm foam mat (Eva foam) is used to place and fixate the FitLight Trainer units, as illustrated in Figure 1a. Either five or four units are positioned in different constellations and as such are used in four different tests. The distance between two closest units is set to either 25 cm or 50 cm.

Through a dedicated device, the light activation of each unit is programmed to be triggered by direct foot contact and the highest sensitivity is chosen. 

In addition to the FitLight Trainer system, the test’s timings are also measured with a wearable IMU device including a 3D accelerometer produced by ST Microelectronics [18]. The device is positioned on the shoe, above the metatarsal part of the participant’s foot of the active leg, and fixed using adhesive tape, as illustrated in Figure 1b. The sampling frequency is set to *f_s_* = 200 Hz.

### 2.3. Applied Tests

Four designed tests [19] are performed by each individual. All tests are repeated with the right and the left leg. In each test, the participant’s task is to make contact with the FitLight units in the prescribed order, respecting the particular constellation of the units used in the test. 

The test performance is guided by the FitLight unit’s lights. The initial unit that the foot should be in contact with is coloured green. Detected contact between the foot and the initial FitLight unit initiates the beginning of the test. To help the participant perform each test, each unit that is to be touched next is coloured blue. When all defined contacts are detected, the test ends. 

Test 1 comprises motion in the anterior–posterior direction—while the participant is keeping balance on the standing leg, the active leg is moving forward and backward, as illustrated in Figure 2. After making contact with the middle unit, the participant proceeds by moving their leg forward, making contact with the frontmost unit and then backward, touching the initial unit again. This cycle is repeated 10 times, altogether giving *L*_1_ = 21 contacts.

The tests performed with the right and the left leg are denoted with Test 1R and Test 1L, respectively.

Test 2 comprises motion in the medio-lateral direction—while the participants is keeping balance on the standing leg, the active leg is moving side to side, as illustrated in Figure 3. When the test is being performed with the right leg, after making contact with the middle unit, the participant proceeds by making contact with the leftmost unit and then with the rightmost unit. When the left leg is active, the order of the unit contacts changes. For both legs, after the initial contact, the unit in the middle is skipped. The cycle is repeated 10 times, altogether giving *L*_2_ = 22 contacts.

The tests performed with the right and the left leg are denoted with Test 2R and Test 2L, respectively.

Test 3 comprises motion following the shape of a lateral triangle—while the participant is keeping balance on the standing leg, the active leg is moving, as illustrated in Figure 4. When the right leg is active, after making contact with the middle unit, the participant proceeds with a forward motion, touching the frontmost unit and then with backward diagonal motion, touching the rightmost unit. When the left leg is active, after touching the frontmost unit, the backward diagonal motion continues, touching the leftmost unit. The cycle is repeated five times, altogether giving *L*_3_ = 16 contacts.

The tests performed with the right and the left leg are denoted with Test 3R and Test 3L, respectively.

Test 4 comprises motion following the shape of a front triangle—while the participant is keeping balance on the standing leg, the active leg is moving, as illustrated in Figure 5. When the right leg is active, after making contact with the middle unit, the participant proceeds by making contact with the units in the following order: the leftmost unit, the middle front, the rightmost unit, the middle front, the leftmost unit. This cycle is repeated five times, altogether giving *L*_4_ = 22 contacts. When the left leg is active, the order of the unit contacts after the initial one is as follows: the rightmost unit, the middle front, the leftmost unit, the middle front, the rightmost unit.

The tests performed with the right and the left leg are denoted with Test 4R and Test 4L, respectively.

### 2.4. Procedure

All participants were familiarized with the tests. After obtaining an oral explanation of the expected test performance, each participant repeated two rehearsals for each of the four tests, with both legs. In total, the familiarization process lasted for approximately 30 min for each individual. 

After familiarization, the participant performed two trials, each including all four tests, performed with both legs, with a pause of at least half an hour between the trials. The order of the tests’ performance as well as the first active leg were chosen randomly. For each test, 42 measurements were obtained for each of the performing legs.

All measurements were supported with video recordings. The measurements were performed during two sessions and at two different locations, in a sports gym and at the premises of the Faculty of Sports and Physical Education in Belgrade.

### 2.5. Measuring Contact Times

The contact times between the foot and the units are measured and accessed through the FitLight Trainer using the dedicated device. The time difference between the last and the first contact point is used as the final test timing result.

The 3D accelerometer signals obtained from the IMU device are passed through a simple zero-phase low-pass filter with a 50 Hz cutoff frequency before further processing.

To detect the contact points between the foot and the ground using IMU 3D accelerometer signals, we apply a rule-based method as follows. For each time sample *n*, we calculate the 3D acceleration magnitude *a*[*n*]. Denoting the measured acceleration components with *a_x_*, *a_y_*, and *a_z_*, we can write:(1)$$a\left[n\right]=\sqrt{a_{x}^{2}\left[n\right]+a_{y}^{2}\left[n\right]+a_{z}^{2}\left[n\right]}.$$

Considering (1), we define the points of contact between the foot and the FitLight unit as those local maximums of *a*[*n*] that exceed a certain threshold value *a_min_* and are at a minimum *n_min_* distance from other maximums. If any two maximums are closer than *n_min_*, the one with the lowest value is discarded. The threshold value *a_min_* is set to 0.25max{*a*[*n*]}, where the maximum acceleration magnitude is considered for each test and participant separately. For all tests considered and for all participants, the difference between two consecutive contact times is larger than 100 ms. Using this margin value as the minimum time distance at *f_s_* = 200 Hz, we set *n_min_* = 20 samples. 

Following the aforementioned procedure for each test, we extract the first *L* contact points. Since the participants start each test from a still position, we set the first detected contact point as the test start. Limiting the size of the extracted set of foot and ground contact points is necessary since participants tend to inertially continue with motion, performing a couple of additional contacts, once the number of contacts defined per test has already been accomplished. The extracted contact points from the 3D acceleration magnitude for one example measurement are illustrated in Figure 6.

The extracted set of *L* contact points is visually inspected and eventually corrected. Finally, we use the difference between the *L*-th and the first contact point as the total test run time.

### 2.6. Statistical analysis

For each of the tests’ total run times, obtained either using the FitLight Trainer or the IMU wearable device, the following basic descriptive statistical indicators are calculated: the minimum value (Min), the maximum value (Max), the mean value (Mean), the standard error of the mean (SEM), the standard deviation (SD), and the coefficient of variation (cV). The normality of the distribution of the results is determined by the Kolmogorov–Smirnov test.

The agreement of the results obtained with the two considered measurement devices is evaluated using the Pearson’s correlation coefficient (Pearson’s r) as well as the intraclass correlation coefficient (ICC) [20]. 

Mean differences between the tests’ run time results obtained from the measurements with two devices are compared by Student’s *t*-test for dependent samples. 

Discrepancies between the results obtained with both devices, are examined with Bland–Altman diagrams and the root-mean-square error (RMSE) values [21]. A value of *p* < 0.05 is accepted as statistically significant [22]. IBM SPSS Statistics (Version 20) predictive analytics software [23] is used for the data analysis.

## 3. Results

The results of the descriptive statistical analysis are presented in Table 1 for the tests’ run times measured by the FitLight Trainer and IMU sensor device, for all four tests and both active legs. The minimum, maximum, and mean value ranged from 3.35 to 4.61, from 5.01 to 6.77, and from 3.99 to 5.52, respectively. The standard error of the mean, standard deviation, and coefficient of variation ranged from 0.051 to 0.104, from 0.331 to 0.671, and from 6.5 to 12.0, respectively.

Table 2 presents the results of the Pearson’s r values, ranging from 0.940 to 0.988. The table also presents the relative differences (A − B)/A between the results of the tests’ run times obtained with the two devices, for all four tests and both active legs, where the FitLight Trainer device is the reference A and IMU device is B. The values range from −0.06% to 1.16%. Finally, the presented ICC values are between 0.967 and 0.994. 

The Student’s *t*-test results of the mean differences between the sample tests’ total run times and RMSE used as a measure of the differences between the obtained values, for all applied tests measured by the FitLight Trainer and IMU device, are presented in Table 3. The results range from −0.212 to 1.853, and from 0.07 s to 0.21 s, respectively. 

Figure 7 shows the Bland–Altman plots for the tests’ total run time measured by the two devices for all tests considered, including a 95% limits of agreement interval (from −1.96SD to +1.96SD). The obtained bias values range from −0.0076 s to 0.0498 s.

## 4. Discussion

The descriptive statistics show the significant homogeneity of the raw results obtained using both devices (Table 1). Overall, the tests’ run times from the FitLight Trainer and IMU device are in the range of 3.36 s to 6.77 s, and from 3.35 s to 6.70 s, respectively.

According to all recognized statistics, the tests’ total run times measured with the FitLight Trainer and IMU devices show high agreement levels. Regarding the validity of the IMU for measuring the tests’ run times, the results of the Student’s *t*-test (Table 3) show no statistically significant mean difference between the total run times obtained by the FitLight Trainer and IMU device. The high ICC values (0.967 ≤ ICC ≤ 0.994) indicate the high consistency of the results obtained using the two devices (Table 2). This is supported by the relative differences, expressed as a percentage (−0.06% *≤* relative differences *≤* 1.16%), between total test run times obtained using the two devices (Table 2). Furthermore, high Pearson’s correlation coefficient values (0.940 ≤ *r* ≤ 0.988) are found in the comparisons between the devices in all of the applied tests (Table 2). 

The Bland–Altman plots also confirm the good agreement between the results obtained using the two devices (Figure 7). The presented results show no statistical significance in the differences between the tests’ run times calculated using the two devices (−0.0076 s ≤ bias ≤ 0.0498 s). Additionally, the small RMSE values (0.07 ≤ RMSE ≤ 0.21) (Table 3) also indicate a high level of agreement between the measurement results of the total run times in the four tests with both legs, and the absence of statistical significance of the differences between the two applied devices. 

Regarding the test–retest reliability of the IMU device for total run time estimation in all four applied tests, the coefficient of variation (6.5% ≤ cV ≤ 12.0%) is in line with the coefficient of variation of the FitLight Trainer device (6.6% ≤ cV ≤ 12.1%). Although the results of the measurements with the two devices are in agreement, based on the coefficient of variation, we can conclude that the repeated measurement gave better results than the first measurement, which indicates that greater familiarization with the tests is needed in order for the tests to be reliable. McMaster et al. [24] show the reliability of an IMU placed on the back of the participant, with a cV of 7.1% being reported.

Moreover, based on the obtained results, we can see that the total run times of Test 2, Test 3, and Test 4 are higher when performed with the non-dominant leg compared to when they are performed with the dominant leg. Test 1 proved to be simple to perform with either the dominant or non-dominant leg. This can be a significant indicator of which movements should be practiced in particular, especially for dancers, for whom the use of all body parts is very important [25]. 

Furthermore, the data presented in this research are in line with [26] and suggest that the sensor locations and their combinations should be guided by the joints of interest and the nature of the movements.

Although the results of the total tests times for all four tests are consistent, measurements with the FitLight Trainer device as a closed system, over which the user has no influence, proved difficult in some cases. Sensitivity to various interferences, such as not responding to certain footwear, contamination of the wireless units leading to a delayed signal, and distance of 25 cm between the units has shown to be too small for the timing measurements of some movements for dancers with more speed in their execution. Additionally, the contact with the units was sometimes too light to be registered, despite the sensitivity being set to the highest level. On the other hand, some contacts were only registered because of the foot passing over the unit, without actually touching it. 

The presented results show that the IMU device can replace the FitLight Trainer to measure test timings. The IMU is more portable, lighter, and easier to use. Unlike the FitLight trainer, it is not sensitive to visibility interferences and can be used in broad physical setups, both indoors and outdoors. Further, the IMU device has the potential to provide a more extensive set of measurement parameters than the FitLight Trainer.

The main drawback of estimating time with the wireless IMU device lies in the wireless transmission. As some packets become lost, the accuracy of the peak value estimation is compromised. In this research, we were careful to conduct measurements in time intervals with high-quality wireless transmission. A protocol for solving this problem in general is under development and will be used in the future.

In addition, considering the demonstrated reliability of using an IMU device for contact time estimation, more parameters will be extracted in the future and the methodology will be appropriately supplemented. In particular, the consistency of the tempo of foot contact points will be investigated together with specific acceleration and angular velocity values. In addition, motion will be analyzed in 3D in detail to provide a comprehensive performance evaluation with respect to specific motion patterns and tests.

## 5. Conclusions

The motivation of this research is inspired by the need for a better understanding of the application of assistive technologies in everyday practice as an aid and a useful tool for trainers in real sports circumstances.

As an alternative to other devices, the IMU device proved to provide accurate timing measurements in lower limb complex movement tests. The small size of the IMU-based systems, and the possible monitoring of the use of multiple sensors on different parts of the body at the same time are the main advantages. Not affecting the athletes’ performance is in line with the fact that assistive technology should monitor activity without interfering with or altering performance. 

In addition, the ability to track movement as well as adjust movement depending on the speed of not only the tempo of the music, but also the ability to accelerate and decelerate within the same form, phrase, or pattern without disturbing the technique, can be very useful for dancers, not only to differentiate between good and less good, but also as a useful tool for development and practice and could be a continuation of this research. 

## Figures and Tables

**Figure 1 sensors-23-01456-f001:**
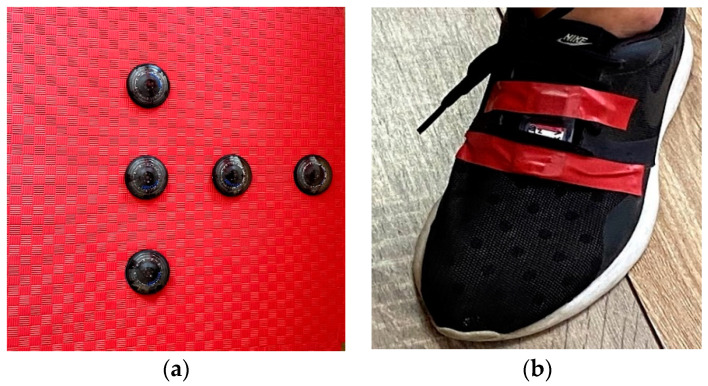
(**a**) FitLight Trainer units positioned and fixed onto foam mat; (**b**) IMU sensor position and orientation on the participant’s shoe.

**Figure 2 sensors-23-01456-f002:**
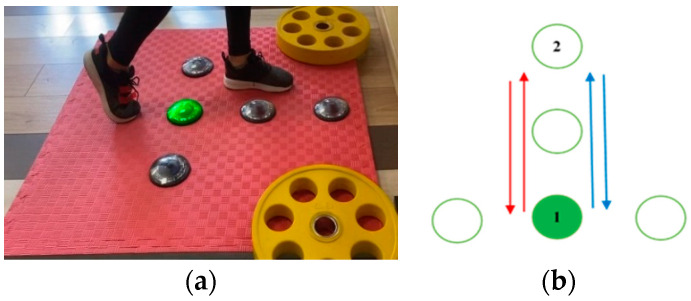
Test 1 (motion in the anterior–posterior direction): (**a**) starting position for right leg execution; (**b**) movement pattern.

**Figure 3 sensors-23-01456-f003:**
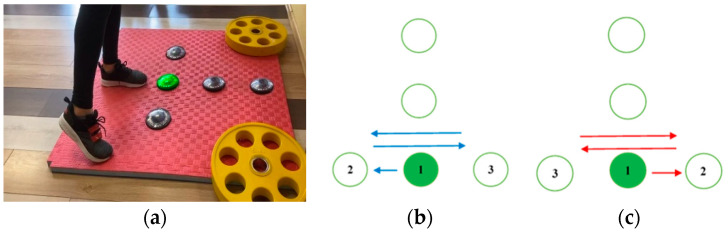
Test 2 (motion in the medio-lateral direction): (**a**) starting position for the right leg execution; (**b**) movement pattern of the active right leg; (**c**) movement pattern of the active left leg.

**Figure 4 sensors-23-01456-f004:**
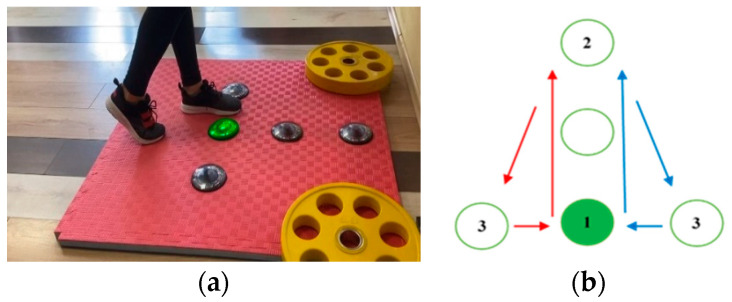
Test 3 (motion following the shape of a lateral triangle): (**a**) starting position for the right leg execution; (**b**) movement pattern.

**Figure 5 sensors-23-01456-f005:**
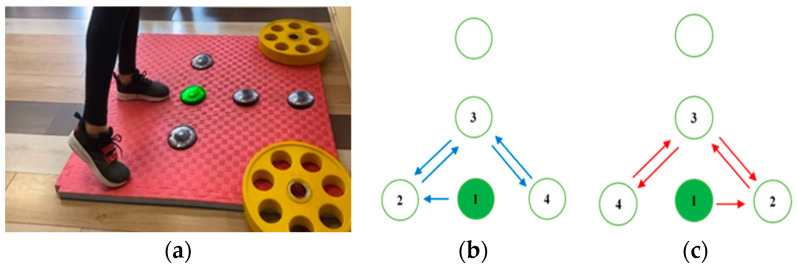
Test 4 (motion following the shape of a front triangle): (**a**) starting position for the right leg execution; (**b**) movement pattern of the active right leg; (**c**) movement pattern of the active left leg.

**Figure 6 sensors-23-01456-f006:**
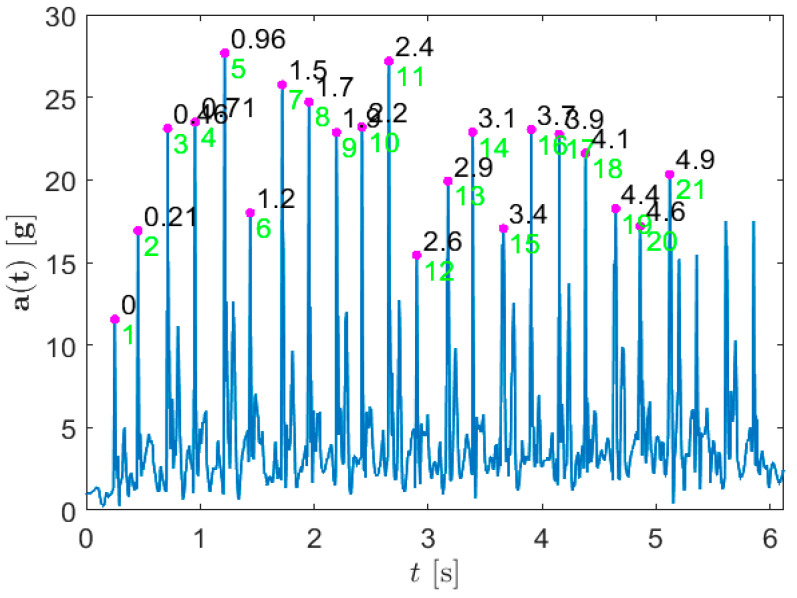
Detecting foot contact point and measuring test’s timing from the IMU acceleration signals. Magenta dots denote the detected points of contact, and in green is the successive contact point enumeration, above which the contact timings with respect to the first contact (in seconds) are also indicated.

**Figure 7 sensors-23-01456-f007:**
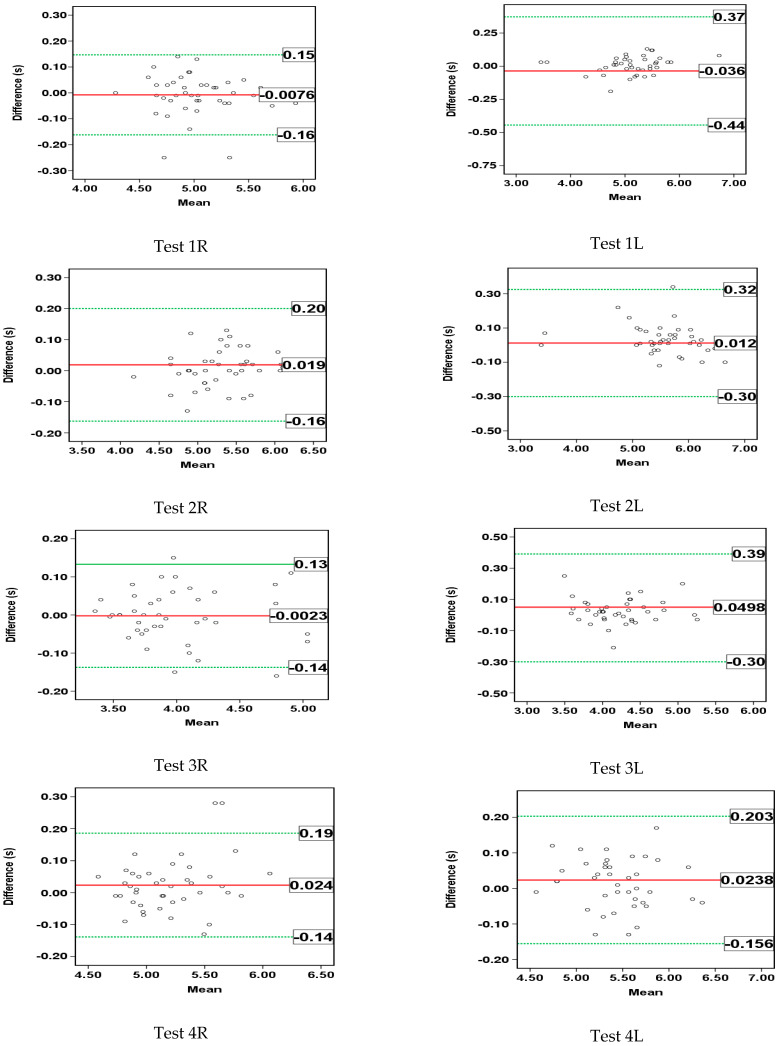
Limits of agreement levels according to the Bland–Altman method for the estimated tests’ run times between FitLight Trainer and IMU device for all four tests and both active legs.

**Table 1 sensors-23-01456-t001:** Descriptive statistics ^1^ for all four tests and both legs (*N* = 42 for each test).

Test	Device	Min (s)	Max (s)	Mean (s)	SEM (s)	SD (s)	cV (%)
Test 1R	FitLight IMU	4.28 4.28	5.91 5.95	5.02 5.03	0.051 0.052	0.331 0.339	6.6 6.7
Test 2R	FitLight IMU	3.94 3.47	6.11 6.10	5.27 5.25	0.075 0.079	0.486 0.512	9.2 9.7
Test 3R	FitLight IMU	3.36 3.35	5.01 5.07	3.99 3.99	0.068 0.069	0.439 0.445	11.0 11.1
Test 4R	FitLight IMU	4.61 4.56	6.09 6.03	5.21 5.18	0.055 0.052	0.355 0.336	6.8 6.5
Test 1L	FitLight IMU	3.47 3.44	6.77 6.69	5.08 5.12	0.094 0.085	0.608 0.552	12.0 10.8
Test 2L	FitLight IMU	3.37 3.37	6.60 6.70	5.52 5.51	0.104 0.102	0.671 0.659	12.1 12.0
Test 3L	FitLight IMU	3.59 3.37	5.87 5.84	4.30 4.25	0.080 0.076	0.517 0.495	12.0 11.7
Test 4L	FitLight IMU	4.56 4.57	6.68 6.66	5.50 5.47	0.070 0.069	0.451 0.445	8.2 8.1

^1^ Including the minimum value (Min), the maximum value (Max), the mean value (Mean), the standard error of the mean (SEM), the standard deviation (SD), and the coefficient of variation (cV).

**Table 2 sensors-23-01456-t002:** Pearson’s correlation coefficients *r*, relative differences, and interclass correlation coefficients ICC for tests run times measured by the FitLight Trainer and IMU device for all four tests and both active legs.

Test	Pearson’s r ^1^	Relative Difference (%)	ICC	Lower–Upper Bound
Test 1R	0.973	−0.41	0.986	0.974–0.992
Test 2R	0.984	0.36	0.991	0.984–0.995
Test 3R	0.988	−0.06	0.994	0.989–0.997
Test 4R	0.973	0.45	0.985	0.973–0.992
Test 1L	0.940	−0.71	0.967	0.938–0.982
Test 2L	0.971	0.22	0.985	0.973–0.992
Test 3L	0.942	1.16	0.970	0.943–0.984
Test 4L	0.979	0.43	0.989	0.980–0.994

^1^*p* < 0.001.

**Table 3 sensors-23-01456-t003:** Student’s *t*-test results for dependent sample and RMSE values for test run times measured by the FitLight Trainer and IMU device.

Test	Student’s *t*-Test Score	*p*-Value	RMSE (s)
Test 1R	−0.950	0.348	0.08
Test 2R	1.316	0.196	0.09
Test 3R	−0.212	0.833	0.07
Test 4R	1.846	0.072	0.08
Test 1L	−1.119	0.270	0.21
Test 2L	0.493	0.625	0.16
Test 3L	1.853	0.071	0.18
Test 4L	1.686	0.099	0.09

## Data Availability

The data presented in this research are available on request from the corresponding author.

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
