# Peer review of "Timing in Lower Limb Complex Movement Tests for DanceSport Athletes: Relation between FitLight Trainer and IMU Measurements"

_sensors, 2023, doi:10.3390/s23031456_

Round 1
Reviewer 1 Report
Dear,
research, data analysis, and interpretation are appropriate. Nicely done.
All comments that are technical in nature are given as follows:
(Chapter 1. Introduction)
Part of the sentence in lines 29-30 should be changed from "Additionally, the authors in [1] find..." to "Additionally, Twitchett et al. [1] find...".
Part of the sentence in line 45 should be changed from "The authors of the review paper [3] show that..." to "Erlikh et al. [3] show that...".
Part of the sentence in line 49 should be changed from "In [4] the authors present a brief..." to "Pustišek et al. [4] present a brief...".
Part of the sentence in line 55 should be changed from "Authors in [5] show..." to "Marković et al. [5] show...".
Part of the sentence in line 59 should be changed from "In [7] the authors present a method..." to "Marković et al. [7] present a method...".
Part of the sentence in line 65 should be changed from "In [8] the authors present a design of..." to "Aylward et al. [8] present a design of..." .
Part of the sentence in line 71 should be changed from "Authors in [11] present a method for..." to "Stančin & Tomažič [11] present a method for...".
Part of the sentence in lines 74 and 75 should be changed from "In [12,13], Kinect-based systems allow extraction and estimation of movement beat alignment using a dance video clip as input." to "Kinect-based systems allow extraction and estimation of movement beat alignment using a dance video clip as input [12,13]".
Part of the sentence in line 81 should be changed from "Using a FitLight Trainer, in [15] the authors examine..." to "Using a FitLight Trainer, Rauter et al. [15] examine...".
Authors should remain consistent with the term "FitLight Trainer" and should correct "FitLight device" (line 93) in "FitLine Trainer" (the word "device" can follow, too).
(Chapter 2. Material and Methods)
Since Pearson's correlation coefficient and Student's test were performed, it would not be bad if authors confirm with one new sentence in line 241, that the data are normally distributed (by Kolmogorov's Smirnov's test for example, without adding new tables).
(Chapter 3. Results)
For the relative differences mentioned in line 255, it is not stated how they were obtained, i.e. to which reference value are they relative, RelDiff = (A-B)/B or RelDiff = (A-B)/A?
I assume that A and B represent these two measurements, performed by FitLight Trainer and IMU, but the question is which of those values should be the reference?... that is, these percentages are relative to which measurement?
Perhaps it would be better to transfer these values from table 2 to table 3, because the values are essentially the same.
Concerning Table 2 as marked below, I would delete "<0.001" and leave only "p" or "p-value".
Concerning Table 2 as marked below, this entire column is the same as the column with Student's t-test significance in Table 3. I would suggest deleting these significances. I assume they refer to these relative differences, but it is not stated or explained anywhere which test was used to obtain these significances.
Finally, concerning Table 2 as marked below, "p<0.001" should be deleted. There is no mention of significance in this column.
Concerning Figure 7 presented below, this black line and the R2 Liner I find unnecessary on these Bland Altman charts... and they do not appear or comment anywhere in the paper. Maybe it's better to get them off the charts.
(Chapter 4. Discussion)
Within the sentence in line 291, the error „wo“ should be changed/corrected into „two“ devices (Table 2).

Author Response
Dear Reviewer,
Thank you for your valuable and constructive comments, we highly appreciate your time and effort. We have made appropriate changes of the resubmitted manuscript in order to address your concerns.
„(Chapter 1. Introduction)
Part of the sentence in lines 29-30 should be changed from "Additionally, the authors in [1] find..." to "Additionally, Twitchett et al. [1] find...".
Part of the sentence in line 45 should be changed from "The authors of the review paper [3] show that..." to "Erlikh et al. [3] show that...".
Part of the sentence in line 49 should be changed from "In [4] the authors present a brief..." to "Pustišek et al. [4] present a brief...".
Part of the sentence in line 55 should be changed from "Authors in [5] show..." to "Marković et al. [5] show...".
Part of the sentence in line 59 should be changed from "In [7] the authors present a method..." to "Marković et al. [7] present a method...".
Part of the sentence in line 65 should be changed from "In [8] the authors present a design of..." to "Aylward et al. [8] present a design of..." .
Part of the sentence in line 71 should be changed from "Authors in [11] present a method for..." to "Stančin & Tomažič [11] present a method for...".
Part of the sentence in lines 74 and 75 should be changed from "In [12,13], Kinect-based systems allow extraction and estimation of movement beat alignment using a dance video clip as input." to "Kinect-based systems allow extraction and estimation of movement beat alignment using a dance video clip as input [12,13]".
Part of the sentence in line 81 should be changed from "Using a FitLight Trainer, in [15] the authors examine..." to "Using a FitLight Trainer, Rauter et al. [15] examine...".
Authors should remain consistent with the term "FitLight Trainer" and should correct "FitLight device" (line 93) in "FitLine Trainer" (the word "device" can follow, too).“
We have corrected all aforementioned comments as suggested. Furthermore, we have corrected part of the sentence, of the same technical nature, in line 314 to “McMaster et al. [25] show…“ in the Discussion section.
(Chapter 2. Material and Methods)
Since Pearson's correlation coefficient and Student's test were performed, it would not be bad if authors confirm with one new sentence in line 241, that the data are normally distributed (by Kolmogorov's Smirnov's test for example, without adding new tables).
The sentence “The normality of the distribution of the results is determined by Kolmogorov-Smirnov test.” is inserted in a new sentence in line 244.
(Chapter 3. Results)
For the relative differences mentioned in line 255, it is not stated how they were obtained, i.e. to which reference value are they relative, RelDiff = (A-B)/B or RelDiff = (A-B)/A?
I assume that A and B represent these two measurements, performed by FitLight Trainer and IMU, but the question is which of those values should be the reference?... that is, these percentages are relative to which measurement?
We thank the reviewer for pointing this out. We have inserted the following detailed explanation in lines 267-269: “In Table 2, the relative differences (A-B)/A) between the results of tests’ run times obtained with two devices, for all four tests and both active legs, where the FitLight Trainer device is the reference A IMU device is B.
Perhaps it would be better to transfer these values from table 2 to table 3, because the values are essentially the same.
Concerning Table 2 as marked below, I would delete "<0.001" and leave only "p" or "p-value".
Concerning Table 2 as marked below, this entire column is the same as the column with Student's t-test significance in Table 3. I would suggest deleting these significances. I assume they refer to these relative differences, but it is not stated or explained anywhere which test was used to obtain these significances.
Finally, concerning Table 2 as marked below, "p<0.001" should be deleted. There is no mention of significance in this column.
We thank the reviewer for pointing this out. Tables 2 and 3 have been modified as suggested.
Concerning Figure 7 presented below, this black line and the R2 Liner I find unnecessary on these Bland Altman charts... and they do not appear or comment anywhere in the paper. Maybe it's better to get them off the charts.
The charts have been corrected and “R2” is removed from the charts.
(Chapter 4. Discussion)
Within the sentence in line 291, the error „wo“ should be changed/corrected into „two“ devices (Table 2).
The error is corrected.

Reviewer 2 Report
Dear Authors,
Thank you very much for sending the article titled: " Timing in Lower Limb Complex Movement Tests for DanceSport Athletes: Relation Between FitLight Trainer and IMU Measurements" for review. The approach of the study appears very original. Especially, it is interesting to check reliability and validity by applying IMU sensor when performing FitLight trainer. The minor revision comments are suggesting as a below:
1) Line 77, It is very appropriate to present the reference [15] for the description of the FitLight Trainer. However, it would be nice to add Specifications such as FitLight size and hardware configuration to enhance readers' comprehension.
2) Line 204, The IMU must have a moving artifact. There is a question as to whether filtering was used for post-processing of the measurement data before calculating the RMS value. If you applied filtering for signal processing, please add an explanation.
3) Line 250~271. At the result, it seems better to change the sentence to a description of the result rather than a description of the table (Line251,252, Line 255~260). For example, in Table 1, Test xx ranged from a few seconds to a few seconds, and the CV was maximum at Test xx at xx seconds and minimum at Test xx at xx seconds.
4) Line 254, Under the “Table1”, add a full name description for the abbreviations used in the table, such as Min, Max, SEM, SD, cV, etc., in the form of comments. And if SD is standard deviation, shouldn't a “±” be prepended?
5) Line 262, At “Table 2”, It seems that it is a common expression to add the p value of Pearson's r as the explanation below in the table, and to mark the value of r with “*”. ex) *: p<0.01. And is p<0.001 correct? Usually 0.01 is used for 99% confidence.
6) Line 266, At “Table 3”, What are the Student’s T-test values in the table? Generally, when presenting the results of the T-test, the average value of the two comparative values (FitLight and IMU in this study) with standard deviation is presented.
7) Line 279, The Bland-Altman plot is a data plotting method used to analyze the agreement between two different analyses. Add the result of the plot as a sentence.
8) The limitations of this study should be added at the end of the “Discussion”. The limitations of this study, I think, are whether the movements of the experiment participants were limited to the range of motion actually measured, the movements were defined with an arbitrary threshold during Acceleration signal processing, and for more accurate analysis, movements such as flexion and extension of the lower-limb should be observed together, but there are things that were not.
9) I expect that more detailed and specific analysis and analysis methods other than simple “peak detection” will be applied in future studies.

Author Response
Dear Reviewer,
Thank you for your valuable and constructive comments, we highly appreciate your time and effort. We have made appropriate changes of the resubmitted manuscript in order to address your concerns.
1) Line 77, It is very appropriate to present the reference [15] for the description of the FitLight Trainer. However, it would be nice to add Specifications such as FitLight size and hardware configuration to enhance readers' comprehension.
We have added more information regarding the FitLight Trainer device in lines 79-81: “FitLight Trainer (Sport Corp. Ontario, Canada) [15] is a commercial system relying on eight led lights, touch and motion sensors. All are integrated into wireless units 10 cm in diameter and weighting 0.3 kg. These units are controlled through a dedicated tablet device with a wireless range of 75 meters and are used to guide an athlete during his or her training session.”
2) Line 204, The IMU must have a moving artifact. There is a question as to whether filtering was used for post-processing of the measurement data before calculating the RMS value. If you applied filtering for signal processing, please add an explanation.
In the presented study, of all contained IMU sensors, only the 3D accelerometer was used. Speed and position, most prone to integration errors and motion artefacts, were not considered. The tests performed showed that the 3D accelerometer is not as sensitive to measurement drift as is the 3D gyroscope, especially in the context of peak value extraction. For this reason, only simple zero-phase lowpass filtering with 50 Hz cutoff frequency was considered. This information is now included in lines 209-211 of the manuscript.
In addition, before further use and RMS calculation, all extracted peaks were visually inspected.
3) Line 250~271. At the result, it seems better to change the sentence to a description of the result rather than a description of the table (Line251,252, Line 255~260). For example, in Table 1, Test xx ranged from a few seconds to a few seconds, and the CV was maximum at Test xx at xx seconds and minimum at Test xx at xx seconds.
We have supplemented the description of the results related to Table 1, Table 2 and Table 3 in Page 7 and 8.
4) Line 254, Under the “Table1”, add a full name description for the abbreviations used in the table, such as Min, Max, SEM, SD, cV, etc., in the form of comments. And if SD is standard deviation, shouldn't a “±” be prepended?
We have added below the Table 1 a footer with a full name description for the abbreviations used in the Table 1.
SD is presented without “±”, because it is represented by itself and not together with another quantity.
5) Line 262, At “Table 2”, It seems that it is a common expression to add the p value of Pearson's r as the explanation below in the table, and to mark the value of r with “*”. ex) *: p<0.01. And is p<0.001 correct? Usually 0.01 is used for 99% confidence.
We thank the reviewer for pointing this out. We have deleted the entire “p” column and added the p values below the table. The significance tests were performed for p<0.001.
6) Line 266, At “Table 3”, What are the Student’s T-test values in the table? Generally, when presenting the results of the T-test, the average value of the two comparative values (FitLight and IMU in this study) with standard deviation is presented.
The Student’s t-test values in Table 3 are “t” scores as is now indicated in the title of Table 3. The t-score is calculated using the standard t-test formula and representing the ratio between the difference between two groups and the difference within the groups (considering the average values and standard deviation). The p value next to the calculated t-score (presented without SD) indicates the probability that the results from our sample data occurred by chance.
7) Line 279, The Bland-Altman plot is a data plotting method used to analyze the agreement between two different analyses. Add the result of the plot as a sentence.
We have added the results of the plot in line 283.
8) The limitations of this study should be added at the end of the “Discussion”. The limitations of this study, I think, are whether the movements of the experiment participants were limited to the range of motion actually measured, the movements were defined with an arbitrary threshold during Acceleration signal processing, and for more accurate analysis, movements such as flexion and extension of the lower-limb should be observed together, but there are things that were not.
The main drawback of estimating time with the wireless IMU device lies in the wireless transmission. As some packets get lost, the accuracy of peak value estimation is compromised. In this study, we were careful to conduct measurements in time intervals with high quality wireless transmission. A protocol for solving this problem is under development and will be used in the future.
This information is now included in the manuscript Discussion section in lines 339-343 of the manuscript.
On the other hand, the distance of 25 cm between the units has shown to be too small for timing measurements with the FitLight Trainer device for some movements and for dancers with more speed in their execution.
In addition, the current methodology developed using IMU signals allows only for contact times estimation. Considering the reliability of the obtained results, methodology upgrade is under development that will in the future allow for estimating a greater range of parameters, besides contact times. Accurate full motion analysis in 3D will allow for a more comprehensive limb position tracking, differentiating flexions from extensions, fully specifying motion characteristics with respect to each move and test performed.
9) I expect that more detailed and specific analysis and analysis methods other than simple “peak detection” will be applied in future studies.
We thank the reviewer for pointing this out. In the future, more parameters, besides contact times, will be extracted and the methodology will be appropriately supplemented. In particular, the consistency of the tempo of foot contact points will be investigated together with specific acceleration and angular velocity values. In addition, motion will be analysed in 3D in detail to give a comprehensive performance evaluation.
This information is now included in the manuscript Discussion section in lines 344-349 of the manuscript.
